# TRAIN THE LATENT, NOT THE IMAGE: JOINT IMAGE COMPRESSION AND STEGANOGRAPHY

## ABSTRACT

Image steganography is the process of hiding secret information in an image through imperceptible changes. Most of recent works hide message in the image by modifying the pixels of image itself. However, those images with hidden messages are not robust to compression such as JPEG, which is used almost everywhere. In order to achieve the ability to compress the image while still having the ability to carry the message, we propose an innovative optimization method which leverages a semi-amortized approach to directly manipulate latent space data for the joint optimization of image compression and steganography. In the compression module, we investigate two of the most popular models in learned image compression with different pre-trained quality: the hyperprior model and the ELIC model. For the steganography module, our method employs the pre-trained fixed neural network steganography (FNNS) model. We compare our method with two state-of-the-art methods such as FNNS-JPEG and LISO-JPEG, achieving significant image compression while maintaining high fidelity and ensuring the accuracy of content upon decoding. The results demonstrate the effectiveness and superiority of our approach.

## 1 INTRODUCTION

Image steganography involves the artful concealment of sensitive data, such as audio, imagery, and textual content (Morkel et al., 2005), within a host image through minimal perturbations. In an optimal scenario, the embedded information remains undetectable to all but the intended recipients, who possess the requisite keys for extraction. Despite minor discrepancies between the steganographic and original images, the presence of covert data remains imperceptible to the uninitiated, rendering steganography a valuable asset in applications such as digital watermarking (Wolfgang & Delp, 1996; Shih, 2007) and patent verification (Lu, 2005).

Conventional steganographic practices often rely on pixel-level image statistics, with the Least Significant Bit (LSB) technique being a prime example (Pevný et al., 2010; Holub & Fridrich, 2012b; Holub et al., 2014). This method ingeniously manipulates and embeds information within the LSBs of an image's pixels, capitalizing on the human visual system's relative indifference to minor color variations for the purpose of information obfuscation.

The advent of deep learning has revolutionized steganography with the advent of end-to-end trainable encoder-decoder neural networks (Zhang et al., 2019; Dong et al., 2018; Baluja, 2017). In particular, convolutional neural networks have demonstrated an uncanny ability to discern and exploit the manifold structure of images (Zhang et al., 2019; Baluja, 2017). These sophisticated methods not only produce highly realistic encrypted images, but also facilitate the encoding of substantial information loads, potentially reaching a density of 6 bits per pixel (bpp). However, this increased capacity comes at the cost of a proportional increase in error rates (Reed et al., 1960).

Innovative methodologies have emerged that frame steganography as a constrained optimization problem, harnessing adversarial learning strategies to embed data by introducing subtle, yet deliberate, perturbations within the image (Kishore et al., 2021). Other cutting-edge approaches amalgamate end-to-end neural networks with optimization algorithms, achieving a remarkable 100% accuracy rate while simultaneously generating images of enhanced naturalism (Chen et al., 2023). However, these methods are sensitive to commonly adopted image compression techniques. Even under the 1 bpp condition (which is much less aggressive than real world image compression), the

accuracy of the extracted information is almost lost after the steganographic image is compressed by JPEG (Wallace, 1992). Even if a differentiable JPEG is added in the optimization process to back propagate the gradient, the PSNR metric is bad and unnatural pictures will be produced. Moreover, the compression quality of training and evaluation must be consistent to have high accuracy, which is not in line with actual transmission situation.

The steganographic image is not as close to the original image as possible; if the mse metric is used, it is easy to make the image blurred, adding the perception metric will make the encrypted image more real and effectively improve the robustness in some detection cases. Therefore, in order to improve the visual quality of steganographic images and increase the security of steganography, we add a GAN-like discriminator (Goodfellow et al., 2014; Agustsson et al., 2018) for adversarial training to achieve the objective of improving the subjective quality of images. Previous work has shown that semi-amortized inference (Kim et al., 2018) can be used to improve R-D performance (Johnston et al., 2017; Yang et al., 2020). We intend to use it in our model to edit latent variables to achieve steganography and reconstruction task. We find that this method can flexibly control each trade-off metric. Our contributions are as follows.

- We are the first to propose a joint optimization of compression and steganography, which solves the problem that steganographic images are destroyed due to the compression process, so that encrypted images can convey more effective information in the case of compression.
- Our method consists of a image compression module and a steganography module, both of which use pre-trained models and are independent of the optimization process based on semi-amortized inference, allowing flexible model configurations.
- We exploit the advantages of GAN models in generating images with better subjective quality and introduce discriminator and adversarial training for optimization. The results show that the subjective quality of the images generated by our method is also better than that of existing methods.

## 2 RELATED WORKS

**Steganography** classic steganography operates directly on the spatial of the cover image to encode the message to be hidden. For example, least significant bit (LSB) steganography. This method sequentially embeds the binary representation of the hidden message in one of the RGB channels of the carrier image. Previous work such as Pixel-Value Differencing (PVD) (Wu & Tsai, 2003) uses the difference in pixel dimension between two images. Highly undetectable steganography (HUGO) (Pevný et al., 2010) uses the minimization of a well-defined distortion metric, which quantifies the perceptual and statistical changes introduced by the embedding process. This distortion metric is typically formulated on the basis of an extended state space, capturing both local and global characteristics of the cover medium. For Wavelet Obtained Weights (WOW) (Holub & Fridrich, 2012a), this algorithm assesses the embedding cost of each pixel in an image using a set of directional filters. The core idea of the WOW algorithm is to adaptively embed secret information based on the local texture complexity of the image. With the development of deep learning. HiDDeN (Zhu et al., 2018) has proposed a encoder-decoder framework to hide messages. Hayes & Danezis (2017) and Zhang et al. (2019) use adversarial training to generate better quality steganographic images, and the latter method can hide up to 6 bpp with error rates of about 13-33%. All these approaches are training based methods, which means using a dataset to train a model and test it on other images. Recent years have seen a novel method called learning-to-optimize (Kishore et al., 2021) which inserts an optimization problem for each processed image. The steganographic image is optimized with respect to the outputs of a fixed (random or pre-trained) decoder and encoder and the optimization problem is solved with gradient-based optimizer, such as L-BGFS (Dennis, 1982).

**Learned end-to-end Image Compression** Over the past decade, learning-based image compression has achieved remarkable success. One of the pioneering contributions in this field was made by Johannes Ballé (Ballé et al., 2017), who first proposed an end-to-end learning framework for image compression and use uniform noise estimator and a parametric entropy model to approximate the probability mass function. Then VAE architecture and hyperprior were proposed (Ballé et al., 2018) for further improvement. They use hyperprior parameter $\hat{z}$ to calculate the parameters of the entropy model. Minnen et al. (2018) utilizes spatial masked convolution as context model, which

improves the compression ratio at the cost of high decoding complexity. Then channel-wise context model is proposed for more effecient context modeling (Minnen & Singh, 2020). ELIC (He et al., 2022) adopts a spatial-channel context model with other architecture improvement, achieving a better balance between compression and computational complexity. To make reconstructed images more realistic and natural, GAN based models are used and verified to be successful in image compression (Mentzer et al., 2020), where conditional GAN is used to constrain consistency between decoding image and origin image.

**Semi-amortized Variational Inference and Code Editing** Kim et al. (2018) and Marino et al. (2018) invented semi-amortized variational inference. In traditional Variational Autoencoders (VAEs), a shared inference network, known as the encoder, generates variational parameters for each sample. This approach, referred to as Amortized Variational Inference (AVI), is computationally efficient because the same network is used globally across all samples. However, the shared nature of the inference network can lead to suboptimal variational parameters, as it may not capture the specific characteristics of individual samples accurately. On the other hand, Stochastic Variational Inference (SVI) performs variational inference for each sample individually. While SVI can produce more accurate and sample-specific variational parameters, it is computationally expensive and often impractical for large datasets. The Semi-Amortized Variational Autoencoders is to initialize the variational parameters for each sample like traditional VAE, then update the variational parameters for each sample to get high-quality posterior approximations. Campos et al. (2019) and Yang et al. (2020) introduce this method to learned end-to-end image compression. Training a fully amortized network is the first step and iteratively optimizing latent is the next step. Gao et al. (2022) proposed Code Editing, a new paradigm for continuous variable bitrate neural image compression based on semi-amortized inference. They edit latent directly towards different optimization target, giving neural image compression more flexibility.

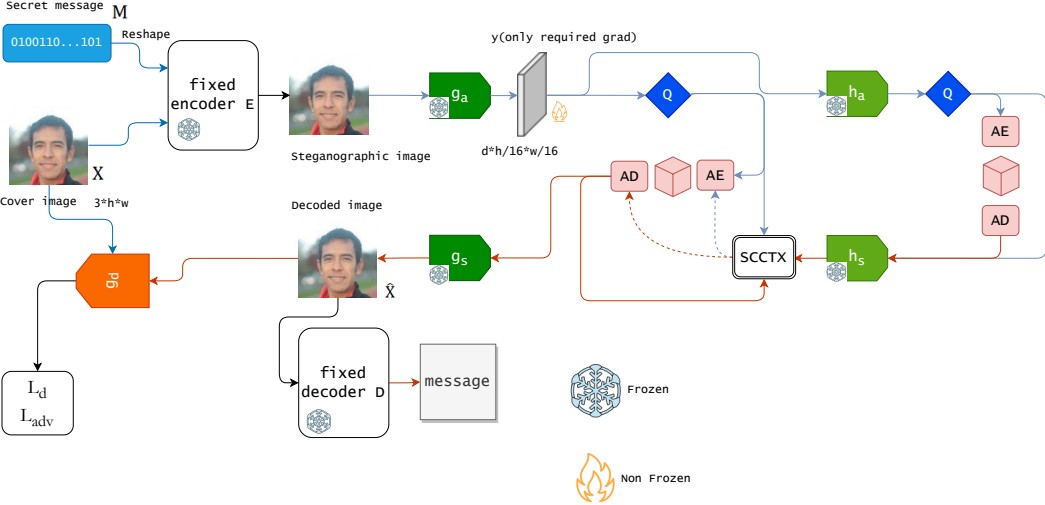

Figure 1: The overall framework of our approach. We use the same architecture of $g_a, g_s, h_a$ and $h_s$ as ELIC (He et al., 2022). SCCTX denotes the spatial-channel context model. We use the uneven 5-group scheme with parallel context models. Fixed encoder and Fixed decoder are several convolutional neural networks with parameters frozen, the same as FNNS (Kishore et al., 2021). There is only $y$ changing during the optimization.

## 3 PROPOSED METHOD

### 3.1 OVERALL FRAMEWORK

Let $X \in [0, 1]^{3 \times H \times W}$ be a color image with height $H$ and width $W$. Let $M \in [0, 1]^{D \times H \times W}$ be a message that we are trying to conceal in $X$, where $D$ specifies the number of bits we need to hide per pixel. We assume that the length of message is $D \times H \times W$. If the message length is not a multiple of $H \times W$ we can simply ignore the unused outputs and view them as zero during optimization.

We assume the involvement of two entities in the information transmission process: the sender and the receiver. The sender uses an encoder network to embed the information within the cover image, producing a steganographic image. In contrast, the receiver utilizes a decoder network to extract information from the target steganographic image. However, considering there is a transmission process, the image needs to be compressed, which means it has information loss during the process. Our primary objective is to guarantee that the receiver can accurately extract the information even after the steganographic image has been subjected to compression and transmission. Furthermore, the steganographic image should closely resemble the original image, both perceptually and quantitatively.

Our structure includes an image compression module and a steganography module, both of which use pre-trained models. The compression and strganography process is shown in Fig. 1. For the compression module, we use both ELIC (He et al., 2022) and Hyper (Ballé et al., 2018) as our coding architecture to simulate the compression and transmission process. Fig. 1 shows its diagram with ELIC. We froze the parameters of all neural networks, retaining only the latent code $y$ as the object that can receive gradients and participate in optimization.

Deep learning-based data compression methods have achieved an increasingly strong performance in visual data compression, outperforming classical codecs such as JPEG and BPG (fabrice bellard, 2015) in terms of rate-distortion performance. Both hyperprior and various context models proposed recently have greatly improved the rate-distortion performance, but optimized reconstruction based on mse loss is easy to generate blurred images, which will also occur with the loss function of ssim (Wang et al., 2004). Some previous work adopted GAN to enhance perceptual quality, such as using a generator and a conditional discriminator to compress images while maintaining subjective quality (Agustsson et al., 2018). In our method we use the discriminator same as HIFIC (Mentzer et al., 2020).

### 3.2 Loss Function

To generate steganographic images with high fidelity, recover messages with low error, and save image in low bpp, the overall optimization object is:

$$L = R + \lambda_1 L_{recon} + \lambda_2 L_{perc} + \lambda_3 L_{adv} + \lambda_4 L_{tv} + \lambda_5 L_{acc}, \tag{1}$$

where the reconstruction loss is mse:

$$MSE = \frac{1}{WH} \sum_{i=0}^{w} \sum_{i=0}^{h} (X_{ij} - \hat{X}_{ij})^2, \tag{2}$$

Perceptual loss $L_{perc}$ is LPIPS-VGG loss (Zhang et al., 2018). Given a discriminator $g_d$, the adversarial loss (Mentzer et al., 2020) is:

$$L_{adv} = -E \log g_d(\hat{x}, \hat{y}), \tag{3}$$

$L_{tv}$ is total variational loss:

$$L_{tv} = \sum_{i,j} \left( (\hat{x}_{i,j-1} - \hat{x}_{i,j})^2 - (\hat{x}_{i+1,j} - \hat{x}_{i,j})^2 \right)^\beta, \tag{4}$$

$L_{acc}$ is the decoding accuracy using cross entropy loss:

$$L_{acc} = \mathbb{E}_{x \sim p_c} CrossEntropy(D(E(X, M), M)), \tag{5}$$

where $p_c$ is cover image distribution, $D$ is the message decoder, $\varepsilon$ is the cover image encoder, $M$ is the message. The $\lambda_1$ used for reconstruction error are the same as those used in the pre-trained image compression models to ensure effectiveness during joint optimization.

### 3.3 Optimization via Code Editing

We use Code Editing (Gao et al., 2022) for joint optimization of compression and steganography. Specifically, given an origin image $x$, we first initialize the continue latent parameters $y \leftarrow f_{\phi_{\lambda_0}}(x)$. Next, we iteratively optimize $y$ to maximize the objective function $L$ as shown in equation 1. In

other words, we directly edit the code $y$. The decoder and entropy parameters $\theta_{\lambda 0}$ are kept constant during the optimization process.

$$y = arg \min_y L, \tag{6}$$

where $L$ is equation 1. Similar to most learned image compression methods, the challenge is that rounding operation is non-differentiable, the majority works of NIC adopt additive uniform noise (AUN) to relax it (Ballé et al., 2017; Ballé et al., 2018), which is also our method.

## 4 EXPERIMENTAL RESULTS

### 4.1 EXPERIMENTAL SETUP

We validate our approach on three distinct datasets:

1) DIV2K (Agustsson & Timofte, 2017): A widely-used dataset for super-resolution and image reconstruction, consisting of 800 high-quality natural images.

2) MS COCO (Lin et al., 2014): A benchmark data set for object detection commonly employed in the field.

3) CelebA (Liu et al., 2015): A well-known dataset for face recognition tasks.

For both the DIV2K and MS COCO datasets, we cropped the images to the size of $512 \times 512$ pixels. For the CelebA dataset, images are cropped to $192 \times 192$ pixels to align with the image compression module.

In all cases, we use random 100 images from the validation set of each dataset. To simulate the distribution of compressed or encrypted messages, we employed random binary bit strings generated from an independent Bernoulli distribution with a probability parameter $p = 0.5$.

Our approach relies entirely on pre-trained models. In the compression module, we use two of the popular models in learned image compression with different quality: the hyperprior model and the ELIC model. We use the pretrained model from the opensourced hyperprior[1] and ELIC[2] model respectively. For the steganography module, we utilize the pre-trained steganography model, which employs a classic encoder-decoder architecture like FNNS (Kishore et al., 2021) and LISO (Chen et al., 2023). More specifically, we use the pretrained encoder-decoder from FNNS directly. We set the iteration step to 1500 and use the Adam optimizer with a learning rate of 1e-3. To demonstrate the influence of JPEG compression, we perform both direct JPEG compression on the carrier image and analyze the steganography effects after training with differentiable JPEG and corresponding JPEG compression. We call our model Hyperbase and ELICbase steganography model. For every $\lambda$ in equation 1, we set $\lambda_2 = 16, \lambda_3 = 1, \lambda_4 = \frac{1}{244}, \lambda_5 = \frac{1}{48}$. Specially for $\lambda_1$ we use it the same as the pre-trained compression model to maintain the same rate-distortion trade-off as pretraining.

### 4.2 EVALUATION METRICS

Steganography algorithms are evaluated along the amount of data that can be hidden in an image, a.k.a *capacity*. Compression algorithms are evaluated along the length of binary data that represent a compressed image, a.k.a. *rate*. Both are evaluated along the similarity between the cover and steganography image, a.k.a *distortion*, This section describes some metrics in evaluating the performance of our model.

**Reed Solomon Bits per pixel** Measuring the accuracy of the amount of data that is hidden in an image is non-trivial. In the actual process of information transmission, certain error correction techniques are needed to help accurately transmit information to the target location, which is more meaningful. Consider a practical situation where a piece of information is encoded into a binary bit stream, there is a probability that some of the information will be successfully recovered through steganography of the image. Since the location of the error is random, it can only allow you to know the proportion of errors, but not the location of the error. The value of the decoded information is completely meaningless.

---

[1] https://github.com/InterDigitalInc/CompressAI
[2] https://github.com/VincentChandelier/ELiC-ReImplemetation

To accurately estimate the payload fraction of our approach, we resort to Reed-Solomon codes (Reed et al., 1960) following (Zhang et al., 2019). Reed-Solomon error correction codes belong to a class of linear block codes, which can encode a data block of length $k$, generating coded data of length $n$ ($n \geq k$). The average ratio of valid message can be seen as $\frac{k}{n}$ and repair errors are at most $\frac{(n-k)}{2}$. This shows that if there is a steganography algorithm that produces a wrong bit with probability $p$ during use, then we expect that our error-correcting bits should be more than the number of bits that produce the error. We refer to this "average ratio" metric as Reed-Solomon bits per pixel (RSBPP). Given the error probability $p$, there is an inequality according to the above:

$$p \cdot n \leq \frac{n - k}{2} \tag{7}$$

Therefore, the RSBPP of the decoding message is less than or equal to $1 - 2p$. Considering an image can be compressed and saved in binary format, bpp (bit per pixel) is one of the important indicators to judge the effectiveness of compression algorithm, which indicates the number of bits occupied by each pixel. RSBPP rate, which means how much hidden message can be load by cover image (in bit stream format) per bit, can be represented by:

$$RSBPP_{rate} = \frac{RSBPP}{bpp} \tag{8}$$

Thus, we can measure the relative load of the steganography technique. Higher RSBPP rate means we can reliably transmit more secret message with less transmission cost or smaller steganographic image size. In other words, the size of steganographic image is utilized by the secret message more efficiently, more secret message can be delivered under the same bandwidth or steganographic image size constraint. Based on this consideration, we regard RSBPP rate as a more meaningful metric for practical image steganography.

**Peak Signal to Noise Ratio** To measure the quality of the steganography and compression image, we use peak signal-to-noise ratio (PSNR). This metric is widely used to measure image distortions.

**Fréchet Inception Distance** Fréchet Inception Distance (FID) is used to measure the gap between two image distributions. If one image distribution is the training set, and the images generated by the generative model are used to form the other distribution, then the FID metric indicates the overall similarity between the generated images and the original images. We use FID to measure the perceptual quality of the steganographic images.

### 4.3 COMPARISON AND ANALYSIS

We compare our method with FNNS (Kishore et al., 2021) and LISO (Chen et al., 2023), two state-of-the-art methods in image steganography with optimization-based method. In practical industrial scenarios, directly transmitting raw float32 image data is highly inefficient and resource-intensive. Saving images in formats like PNG, which are lossless, can result in large file sizes, making them unsuitable for real-time or bandwidth-constrained environments. Therefore, it is essential to employ various compression techniques to reduce the size of image data to compare these steganography methods. For FNNS, we saved the final steganographic image in fp16, PNG, and different-quality JPEG formats after loading cover image information, to show FNNS's information recovery ability under different saving methods. We found that the higher the bpp (bits per pixel), the higher the accuracy of the information that can be recovered. Additionally, for different qualities of JPEG methods, we incorporate a differentiable JPEG method into our optimization process, which allows the gradient to pass through the parameters during optimization. It makes image steganography more adaptable to JPEG compression.

For LISO, there is one kind of LISO employs an approximate JPEG layer, where the forward pass performs standard JPEG compression and the backward pass is an analytic function. The improved method is called LISO-JPEG (Chen et al., 2023). Since the pre-trained models in LISO-JPEG are not completely released, we were only able to compare LISO-JPEG with our method in DIV2K.

As can be seen in Tables 1, 2, 3, FNNS-png means saving the steganographic image in png format (Portable Network Graphics) and reloading it to measure the decoding message accuracy, similar to "-jpg90","-jpg70". The number after "jpg" means jpeg quality, "_TRAIN" after jpg means adding differentiable jpeg layer in the optimization pipeline, where the gradient can be propagated backward. And different numbers behind the hyperprior and ELIC means different trade-off (different $\lambda$ in R-D loss) between rate and distortion in pre-training process.

Table 1: Steganography and compression results on CelebA dataset

| Dataset | Method | 1bit | | | | |
|---|---|---|---|---|---|---|
| | | PSNR | BPP | Accuracy | RSBPP rate | FID |
| CelebA | FNNS | 27.55 | 96.00 | 100.00% | 0.010 | \ |
| | FNNS-fp16 | 27.5 | 48.00 | 99.8% | 0.031 | \ |
| | FNNS-png | 27.5 | 16.68 | 99.69% | 0.060 | 126.39 |
| | LISO-png | 35.62 | 17.08 | 100.00% | 0.062 | 108.30 |
| | FNNS-jpg90 | 25.63 | 3.77 | 56.79% | 0.036 | 133.14 |
| | FNNS-jpg70 | 25.43 | 2.55 | 54.76% | 0.037 | 133.47 |
| | FNNS-jpg50 | 34.74 | 1.15 | 51.96% | 0.034 | 135.74 |
| | LISO-jpg90 | 32.68 | 2.12 | 55.61% | 0.036 | 72.04 |
| | LISO-jpg70 | 25.43 | 1.03 | 51.86% | 0.036 | 54.76 |
| | LISO-jpg50 | 34.74 | 0.74 | 51.04% | 0.028 | 54.80 |
| | FNNS-JPG50_TRAIN | 17.50 | 2.80 | 68.16% | 0.130 | 229.83 |
| | FNNS-JPG70_TRAIN | 17.54 | 3.84 | 73.02% | 0.120 | 248.98 |
| | FNNS-JPG90_TRAIN | 20.25 | 5.76 | 78.32% | 0.0983 | 204.66 |
| | hyperbase_3 | 25.27 | 1.386 | 66.40% | 0.237 | 212.15 |
| | hyperbase_4 | 25.8 | 1.748 | 71.93% | 0.251 | 178.06 |
| | hyperbase_5 | 26.40 | 1.508 | 74.80% | 0.329 | 155.01 |
| | hyperbase_6 | 26 | 3.115 | 85.69% | 0.2291 | 134.24 |
| | hyperbase_7 | 26.5 | 3.239 | 85.40% | 0.219 | 133.23 |
| | hyperbase_8 | 28.15 | 3.46 | 84.80% | 0.201 | 101.95 |
| | ELICbase_3 | 22.1 | 2.718 | 60.49% | 0.077 | 305.4 |
| | ELICbase_4 | 24.06 | 2.468 | 64.06% | 0.114 | 238.1 |
| | ELICbase_5 | 26.42 | 2.7 | 83.00% | 0.244 | 160.84 |
| | ELICbase_6 | 29.22 | 2.69 | 87.10% | 0.276 | 129.72 |

Table 2: Steganography and compression results on DIV2K dataset

| Dataset | Method | 1bit | | | | |
|---|---|---|---|---|---|---|
| | | PSNR | BPP | Accuracy | RSBPP rate | FID |
| DIV2K | FNNS | 23.04 | 96.00 | 100.00% | 0.010 | \ |
| | FNNS-fp16 | 23.05 | 48.00 | 100.00% | 0.031 | \ |
| | FNNS-png | 23.04 | 21.78 | 99.98% | 0.046 | 208.15 |
| | LISO-png | 33.83 | 17.08 | 100.00% | 0.062 | 30.53 |
| | FNNS-jpg90 | 22.87 | 5.15 | 62.45% | 0.048 | 112.6 |
| | FNNS-jpg70 | 22.76 | 2.86 | 58.40% | 0.059 | 111.26 |
| | FNNS-jpg50 | 22.68 | 1.98 | 56.31% | 0.064 | 117.57 |
| | LISO-jpg90 | 27.44 | 3.35 | 57.40% | 0.044 | 44.14 |
| | LISO-jpg70 | 29.14 | 1.61 | 52.09% | 0.026 | 47.54 |
| | LISO-jpg50 | 28.66 | 1.19 | 50.69% | 0.012 | 52.05 |
| | FNNS-JPG50_TRAIN | 16.543 | 2.944 | 70.75% | 0.141 | 305.85 |
| | FNNS-JPG70_TRAIN | 17.237 | 3.907 | 76.05% | 0.133 | 323.32 |
| | FNNS-JPG90_TRAIN | 20.37 | 5.86 | 66.00% | 0.055 | 165.11 |
| | LISO-JPEG | 15.41 | 4.20 | 99.47% | 0.236 | 292.24 |
| | hyperbase_3 | 23.08 | 1.64 | 74.18% | 0.295 | 118.53 |
| | hyperbase_4 | 23.20 | 1.57 | 80.58% | 0.376 | 123.83 |
| | hyperbase_5 | 23.74 | 1.66 | 81.21% | 0.376 | 114.67 |
| | hyperbase_6 | 23.28 | 3.53 | 91.75% | 0.237 | 104.43 |
| | hyperbase_7 | 23.82 | 3.42 | 92.58% | 0.249 | 95.98 |
| | hyperbase_8 | 24.90 | 3.83 | 92.20% | 0.220 | 84.73 |
| | ELICbase_3 | 19.76 | 2.61 | 68.60% | 0.135 | 202.96 |
| | ELICbase_4 | 21.58 | 2.39 | 68.40% | 0.163 | 167.04 |
| | ELICbase_5 | 24.71 | 2.46 | 84.15% | 0.278 | 111.15 |
| | ELICbase_6 | 27.80 | 2.82 | 88.00% | 0.270 | 82.14 |

Table 3: Steganography and compression results on MSCOCO dataset

| Dataset | Method | 1bit | | | | |
|---|---|---|---|---|---|---|
| | | PSNR | BPP | Acc Rate | RSBPP rate | FID |
| MSCOCO | FNNS | 30.37 | 96.00 | 100.00% | 0.010 | \ |
| | FNNS-fp16 | 30.37 | 48.00 | 100.00% | 0.021 | \ |
| | FNNS-png | 30.36 | 19.03 | 99.95% | 0.052 | 111.16 |
| | LISO-png | 33.83 | 18.12 | 100.00% | 0.054 | 34.65 |
| | FNNS-jpg90 | 29.94 | 3.64 | 57.65% | 0.042 | 109.43 |
| | FNNS-jpg70 | 29.68 | 1.79 | 53.76% | 0.042 | 181.18 |
| | FNNS-jpg50 | 29.40 | 1.25 | 52.51% | 0.040 | 178.09 |
| | LISO-jpg90 | 30.68 | 1.88 | 53.65% | 0.03 | 109.43 |
| | LISO-jpg70 | 29.68 | 1.35 | 51.76% | 0.025 | 181.18 |
| | LISO-jpg50 | 28.46 | 1.08 | 50.30% | 0.022 | 178.09 |
| | FNNS-JPG50_TRAIN | 16.77 | 2.88 | 70.20% | 0.140 | 280.31 |
| | FNNS-JPG70_TRAIN | 17.49 | 3.85 | 74.10% | 0.125 | 227.34 |
| | FNNS-JPG90_TRAIN | 21.00 | 5.53 | 77.30% | 0.099 | 167.48 |
| | hyperbase_3 | 23.49 | 1.40 | 69.60% | 0.280 | 186.99 |
| | hyperbase_4 | 23.14 | 1.51 | 74.25% | 0.322 | 165.04 |
| | hyperbase_5 | 23.19 | 1.564 | 76.48% | 0.339 | 150.29 |
| | hyperbase_6 | 22.01 | 3.58 | 87.69% | 0.211 | 142.31 |
| | hyperbase_7 | 22.20 | 3.59 | 87.30% | 0.208 | 133.84 |
| | hyperbase_8 | 22.72 | 3.92 | 86.54% | 0.186 | 119.06 |
| | ELICbase_3 | 21.56 | 2.032 | 59.60% | 0.094 | 228.12 |
| | ELICbase_4 | 22.67 | 1.963 | 67.60% | 0.179 | 205.49 |
| | ELICbase_5 | 24.95 | 2.344 | 84.60% | 0.295 | 137.21 |
| | ELICbase_6 | 26.89 | 2.735 | 86.75% | 0.269 | 116.55 |

While FNNS and LISO can attain a zero error rate, this comes at the expense of using PNG or other space-consuming formats. When JPEG is employed to compress the steganographic image and then retrieve the information, the recovery rate significantly drops (e.g. FNNS-jpg90 and LISO-jpg90). Even it can adopt an approximate JPEG layer where the forward pass performs normal JPEG compression and the backward pass is an identity function like LISO-JPEG (Chen et al., 2023) or FNNS-JPG90_TRAIN in practical scenarios, the RSBPP rate and image quality measured by FID and PSNR are all much worse than our methods. The RSBPP rate of our method (hyberbase and ELICbase) is substantially higher than that of FNNS and LISO, and enables better tradeoff between image compression and image steganography. These findings highlight the effectiveness and superiority of our approach.

Fig. 2 gives several visual examples. We can observe that when image compression is considered, the image quality of FNNS-jpg90 and FNNS-JPG90_TRAIN is much worse than our methods Hyperbase and ELICbase. Our methods can produce steganographic images which are visually similar to the original cover images. Note that there is a tradeoff between steganographic image quality, compression ratio and steganography accuracy as shown in Table 1, 2, 3 the last column is shown just to give readers a feeling about the image quality of other methods. We do not need to beat it on this single metirc.

## 4.4 STEGANALYSIS

Steganalysis, as a critical component of information security, is dedicated to the detection of covert communications embedded within digital media. The primary objective of steganalysis tools is to ascertain whether an image has been manipulated to conceal a message. This domain encompasses two principal methodologies: statistical steganalysis and neural steganalysis.

Statistical steganalysis relies on the identification of deviations from the expected statistical properties of an image, detecting LSB (least significant bit) steganography of an image (Gupta & Bhushan, 2012). These deviations may indicate the presence of hidden data. A notable example of a statistical steganalysis tool is StegExpose (Boehm, 2014), which integrates a variety of detection algorithms, such as the Chi-square attack (Westfeld & Pfitzmann, 1999b) and RS (Regular/Singular) analy-

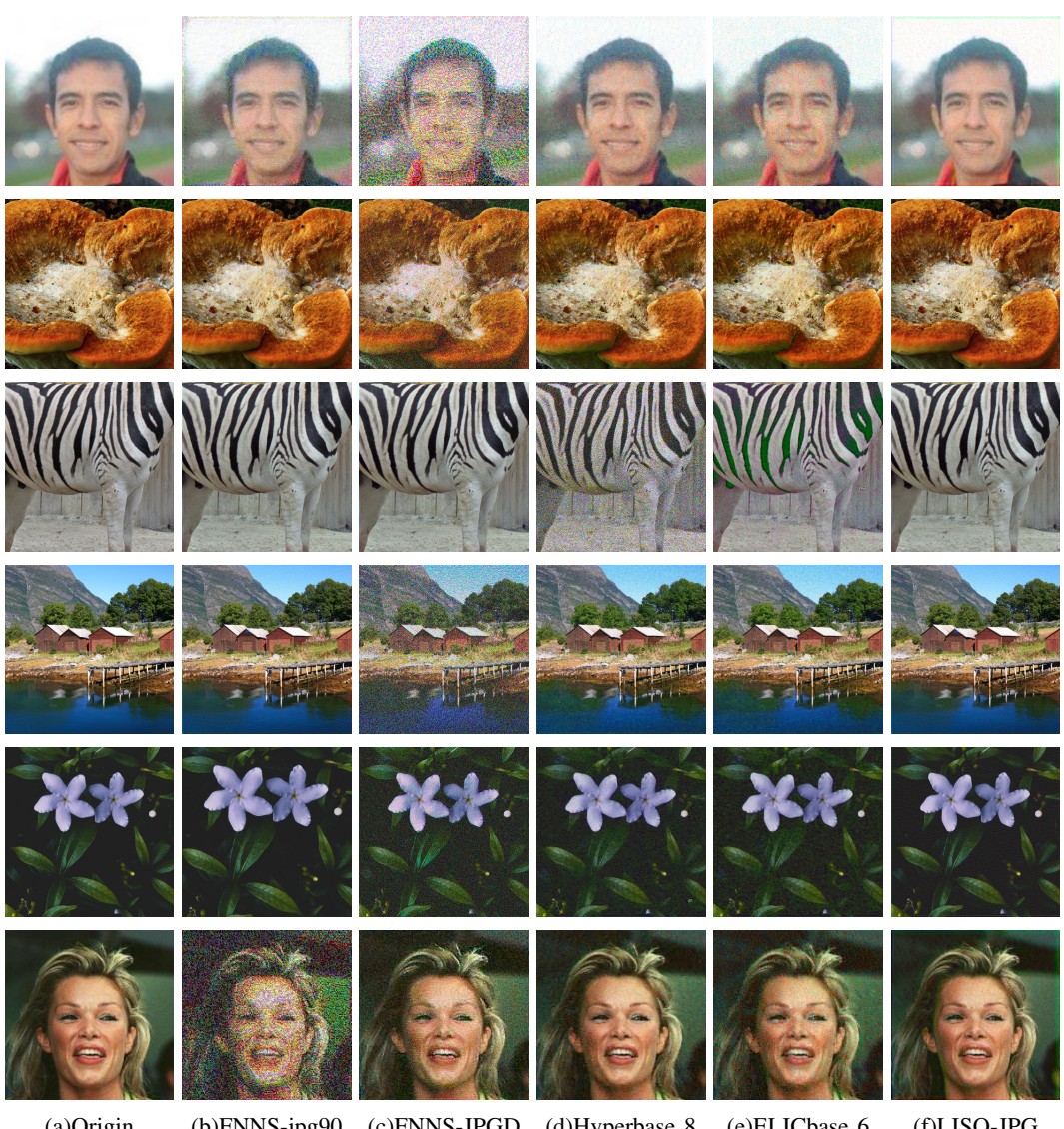

| (a)Origin | (b)FNNS-jpg90 | (c)FNNS-JPGD | (d)Hyperbase_8 | (e)ELICbase_6 | (f)LISO-JPG |

Figure 2: Visual results. FNNS-JPGD means FNNS-jpg90_TRAIN in Table 1, 2, 3

sis (Westfeld & Pfitzmann, 1999a), PrimarySets (Dumitrescu et al., 2002). The Chi-square attack evaluates the distribution of pixel values to detect alterations, while RS analysis focuses on the differences in the statistical behavior of regular and singular pixels to infer the presence of hidden information.

In contrast, neural-based steganalysis leverages the power of neural networks to learn and recognize complex patterns that are indicative of steganographic manipulation (Ye et al., 2017; You et al., 2020). This approach often involves training deep neural networks on large datasets of clean and steganographic images (images containing hidden messages). The trained models can then be used to classify new images with high accuracy, even when the steganographic techniques used are highly sophisticated and subtle. Compared to traditional statistical steganalysis, neural network-based steganalysis exhibits significantly greater power and effectiveness. Neural networks are capable of successfully detecting hidden messages even at low bit-per-pixel (bpp) rates, such as below 0.5 bpp. To evaluate the security of our proposed method, using StegExpose (Boehm, 2014) as the detection tool following FNNS and LISO, we demonstrate its ability to evade detection in Table 4. If image compression is not considered, both FNNS and LISO can achieve zero error rate and LISO can achieve nearly zero detection accuracy. For practical applications, compression is necessary.

Though LISO-jpg90 has low detection accuracy, its error rate is very high. Our methods (Hyperbased and ELICbase) outperform previous methods regarding error rate and detection accuracy.

Table 4: Steganalysis results using images produced by different methods, evaluated on results on 3 datasets, the quality of "jpg" and "diffjpg" is 90, Hyperbase is 8, ELICbase is 6. The error rate, following previous works, is one minus the accuracy in Table 1, 2, 3. Higher detection accuracy means the steganography method is easier to be discovered by detection tool, so lower is better.

| Dataset | Method | Error Rate↓ | PSNR↑ | Detection accuracy↓ |
|---------|--------|-------------|-------|---------------------|
| CelebA | FNNS | 0 | 27.55 | 62% |
| | FNNS-jpg90 | 43.21% | 27.5 | 44% |
| | FNNS-jpg90_TRAIN | 21.68% | 20.25 | 90% |
| | Hyperbase | 15.20% | 28.15 | 36% |
| | ELICbase | 12.90% | 29.22 | 28% |
| | LISO | 0% | 30.15 | 2% |
| | LISO-jpg90 | 44.39% | 32.68 | 14% |
| MSCOCO | FNNS | 0 | 30.37 | 39% |
| | FNNS-jpg90 | 42.35% | 29.94 | 15% |
| | FNNS-jpg90_TRAIN | 22.70% | 21.00 | 61% |
| | Hyperbase | 13.46% | 22.72 | 23% |
| | ELICbase | 13.25% | 26.89 | 7% |
| | LISO | 0% | 30.42 | 1% |
| | LISO-jpg90 | 46.35% | 28.46 | 4% |
| DIV2K | FNNS | 0 | 23.04 | 39% |
| | FNNS-jpg | 37.55% | 23.04 | 15% |
| | FNNS-jpg90_TRAIN | 24.50% | 20.37 | 61% |
| | Hyperbase | 7.80% | 24.9 | 23% |
| | ELICbase | 12% | 27.8 | 7% |
| | LISO | 0.0% | 30.95 | 0% |
| | LISO-jpg90 | 43.60% | 27.44 | 4% |

## 5 CONCLUSION

We propose an innovative optimization method for joint optimization of image compression and steganography. Our method has demonstrated superior efficiency compared to existing techniques such as FNNS-JPEG and LISO-JPEG, achieving significant image compression while maintaining high fidelity and ensuring the accuracy of steganographic content upon decoding. By balancing the image compression rate with the steganographic payload, we have reached a level of performance that is considered state-of-the-art. In the future, our research will continue to explore strategies for reducing compression rates further while maintaining low error rates and enhancing the quality of steganographic images, aiming to push the boundaries of what is achievable in the field of advanced image processing for security and data efficiency.

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

## A APPENDIX

### A.1 DETAIL OF NETWORK ARCHITECTURE AND HYPERPARAMETER

Table 5: Architecture of main part of ELIC.

| Analyzer $g_a$ | Synthesizer $g_s$ |
|---|---|
| in: 3-channel image | in: M-channel symbols |
| Conv $5 \times 5$, s2, N | Attention |
| ResBottleneck×3 | TConv $5 \times 5$, s2, N |
| Conv $5 \times 5$, s2, N | ResBottleneck×3 |
| ResBottleneck×3 | TConv $5 \times 5$, s2, N |
| Attention | Attention |
| Conv $5 \times 5$, s2, N | ResBottleneck×3 |
| ResBottleneck×3 | TConv $5 \times 5$, s2, N |
| Conv $5 \times 5$, s2, M | Attention |
| Attention | TConv $5 \times 5$, s2, 3 |

We show the detail architecture for hyper and ELIC model, for hyper model, it utilizes Generalized Divisive Normalization (GDN) for normalization. GDN is a non-linear normalization technique that is similar to Batch Normalization (BN) but is specifically designed to better capture the statistical

properties of natural images and transform them into a Gaussian distribution. Every $\lambda$ in equation 1, we set $\lambda_2 = 16, \lambda_3 = 1, \lambda_4 = \frac{1}{244}, \lambda_5 = \frac{1}{48}$. Specially for $\lambda_1$ we use it the same as the pre-trained model to maintain the trade-off in pretrain process.

Software and Hardware

PyTorch Version: 2.3.0

Hardware: NVIDIA A100 GPU

Some important libraries:

diffJPEG: We used the diffJPEG library to enable differentiable JPEG compression, allowing us to incorporate JPEG compression directly into the training pipeline.

CompressAI: We utilized the CompressAI open-source library for neural network-based image compression. CompressAI provides a variety of pre-trained models and tools for training custom compression models.

Table 6: Architecture of main part of Hyper.

| Analyzer $g_a$ | Synthesizer $g_s$ |
| --- | --- |
| in: 3-channel image | in: M-channel symbols |
| Conv 5 × 5, s2, N | TConv 5 × 5, s2, N |
| GDN | IGDN |
| Conv 5 × 5, s2, N | TConv 5 × 5, s2, N |
| GDN | IGDN |
| Conv 5 × 5, s2, N | TConv 5 × 5, s2, N |
| GDN | IGDN |
| Conv 5 × 5, s2, M | TConv 5 × 5, s2, 3 |

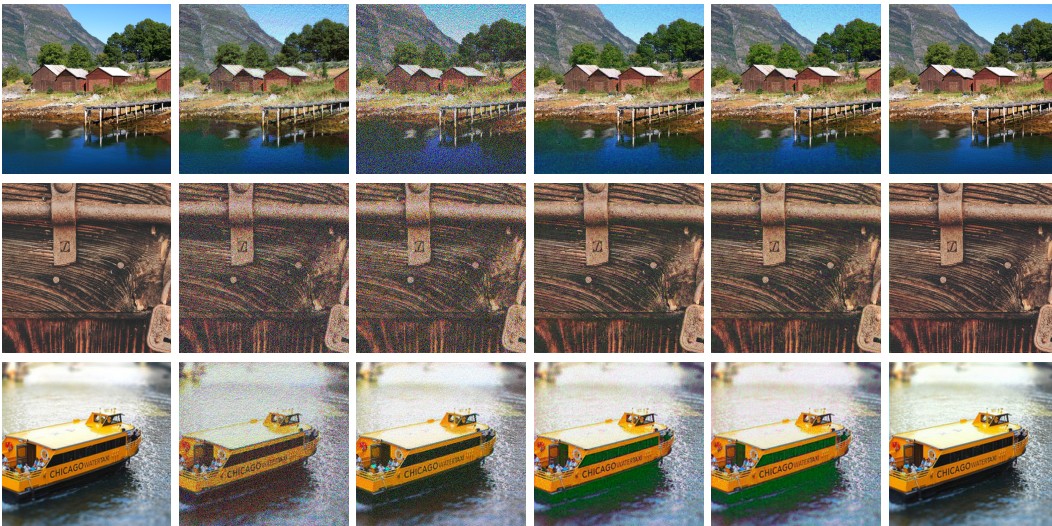

Figure 3: Visual result, JPGD means using differentiable JPEG estimator, the same as FNNS-jpg90-TRAIN

