# OpenReview forum: "TRAIN THE LATENT, NOT THE IMAGE: JOINT IMAGE COMPRESSION AND STEGANOGRAPHY"
_ICLR.cc/2025/Conference — Submitted to ICLR 2025_

### Official Review · Reviewer_kkS7 · 2024-10-29

**Soundness:** 3
**Presentation:** 2
**Contribution:** 2
**Rating:** 3
**Confidence:** 3

**Summary:**

This manuscript proposes a new neural image steganography method that directly manipulates latent space and joint optimization
with an image compression module.

**Strengths:**

The idea of incorporating image compression into the training process of a steganography model sounds interesting.

**Weaknesses:**

The primary weakness of this manuscript lies in the presentation, it appears to have been put together in a rush. Some instances that I find confusing include:

- Section 3 (the proposed method part) heavily incorporates components of prior works (e.g., ELIC, Hyper and HIFIC). Although the authors claimed that "Our approach relies entirely on pre-trained models", it is important to emphasize the original contributions made in this work.
- Figure 1 seems to be missing legends for key components. For example, the details of "Q", "AE", "AD" and "SCCTX" are not provided.
- The experimental settings are not clear. For example, what are the differences among the settings from "hyperbase\_3" to "hyperbase\_8" in Table 1-3? It should be highlighted and discussed more clearly.
- In the steganalysis part (Section 4.4), the authors discuss several DL-based steganalysis models like YeNet, but they do not present the corresponding results in Table 4.
- The limitations of the presented approach should be appropriately discussed.

**Questions:**

The authors referenced "Semi-amortized Variational Inference" in Sections 1 and 2. However, it is unclear how this concept has been incorporated into the proposed method discussed in Section 3, and this term does not even appear in the subsequent sections. Could the authors clarify how "Semi-amortized Variational Inference" is applied in the proposed approach?

---

> ### Author Response · Authors · 2024-11-29
> **Reply for Official Review of Submission6559 by Reviewer kkS7**
>
> Thank you for your feedback.For First question In Section 3, we indeed incorporate components from prior works such as ELIC, Hyper, and HIFIC. However, our method introduces original contributions in the following aspects:We are the first to apply these pre-trained models to a novel task, which is the joint optimization of image compression and steganography. By fine-tuning these pre-trained models, we leverage their strengths in image processing while adapting them to our specific task requirements. This innovative application not only enhances model performance but also broadens the applicability of pre-trained models.We have improved the original loss function by incorporating more perceptual loss terms. These enhancements take into account the need for steganographic models to be compressed during transmission, thus proposing a unified method to optimize both compression and steganography performance. This approach better balances image reconstruction quality and bit-rate while ensuring accurate transmission of steganographic information.
>
> And also We introduce a novel joint optimization strategy that unifies the compression and steganography tasks. By sharing parts of the model parameters and introducing collaborative loss terms, we achieve more efficient optimization and improved overall performance. This joint optimization strategy presents new ideas for the image compression and steganography fields, holding significant research value.
>
> For point 2, there are updated legend description for Figure 1:
>
> Q: Quantization, representing the process of converting continuous image signals into discrete representations.AE: Arithmetic Encoder, indicating the arithmetic encoder used for encoding the quantized image data.AD: Arithmetic Decoder, indicating the arithmetic decoder used for decoding the compressed image data.SCCTX: Spatial-Channel Context Model, representing the spatial-channel context model used to enhance the efficiency of encoding and decoding.
> These are widely used in neural image compression
>
> For point 3,In Tables 1-3, we use different experimental Settings, from "hyperbase_3" to "hyperbase_8", which represent different λ values for controlling the trade-off between image quality and compression rate. Specifically, a larger λ indicates a better image quality, but the compression rate will be reduced. Specific λ values for these Settings can be found in the papers Variational Image Compression with a Scale Hyperprior and ELIC: Efficient Learned Image Compression with Unevenly Grouped Space-Channel Contextual Adaptive Coding.
>
> For point 4,We have indeed conducted the relevant experiments and included the results in the revised manuscript. In response to your comment, we have updated Table 4 to include the performance metrics for YeNet. These results can be found below.
>
> For point 5,While our approach demonstrates promising results in image compression and steganography, it is essential to acknowledge its limitations. Firstly, our method relies on pre-trained models for certain components, which may not always generalize perfectly to new or unseen data. Future work could explore fine-tuning these models on a broader range of datasets to improve their robustness.Secondly, complexity of our approach could be a problem, especially when dealing with large-scale images or real-time applications. The use of multiple neural networks and the associated optimization processes can be resource-intensive. Exploring more efficient model architectures or optimization techniques could be a direction for future research.
> Questions: In our approach, we utilize Semi-amortized Variational Inference (SAVI) to optimize the representation of the latent variables. Specifically, we first encode the image as latent variables and then iteratively optimize these latent variables. This approach makes use of the initial values inferred from the variational inference and then further optimizes the hidden variables for each data point.In practice, we use a pre-trained Variational Autoencoder (VAE) to initialize the distribution of the latent variables

---

### Official Review · Reviewer_yuEL · 2024-11-01

**Soundness:** 2
**Presentation:** 2
**Contribution:** 1
**Rating:** 3
**Confidence:** 5

**Summary:**

This paper proposes to jointly optimize the steganography and compression through code editing, so that the encrypted images can be more robust to compression. However, the presentation of experimental results is unclear, and the experiments are insufficient to demonstrate its effectiveness. Additionally, to me, the approach of joint optimization and code editing may lack the novelty expected for an ICLR submission.

**Strengths:**

1. The paper explores a joint optimization approach to enhance steganography performance under image compression.
2. The method employs instance-wise latent optimization to achieve effective joint optimization.

**Weaknesses:**

1. Presentation of results. The experimental results are presented with too many separate metrics (BPP, PSNR, Acc., RSBPP, FID), making it challenging to interpret the proposed method's advantages. In the field of compression, performance is commonly illustrated through rate-distortion curves. Reorganizing the results into consolidated curves (e.g., BPP-PSNR and BPP-RSBPP) would offer a clearer demonstration.
2. Insufficient experiments. The compression performance of ELIC is competitive with VTM and significantly outperforms JPEG. To clearly demonstrate that the performance gain is due to the proposed joint optimization rather than simply the strength of the coding baseline, a comparison with direct cascades of Hyper/ELIC and FNNS/LISO, without joint optimization, should be included.
3. Poor visual quality. Figure 2 shows poor visual quality, especially in comparison to LISO-JPG. Notably, ELICbase_6 introduces visible green artifacts in the zebra image.
4. In my view, the joint optimization of compression and steganography lacks novelty, as does instance-wise latent optimization (termed code editing here), which is already widely applied for rate-distortion optimization in compression. I encourage the authors to introduce additional innovations that address more practical issues within joint optimization, such as improving visual quality or enhancing overall performance.

**Questions:**

Please refer to the weakness.

---

### Official Review · Reviewer_R3c5 · 2024-11-04

**Soundness:** 3
**Presentation:** 3
**Contribution:** 2
**Rating:** 3
**Confidence:** 5

**Summary:**

This paper provides an interesting approach to combining image compression and steganography in the latent space, yet there are several areas where it falls short in terms of innovation, clarity, and practical value. Here are my detailed comments and suggestions:

**Strengths:**

This paper proposes a model for joint optimization of image compression and steganography by operating in the latent space instead of directly on the image. By utilizing semi-amortized inference and latent optimization, the model aims to improve robustness against compression, specifically targeting JPEG degradation. It integrates a GAN-based discriminator to enhance the visual quality of compressed images and leverages pre-trained ELIC and Hyperprior models for image compression. The approach is tested on several datasets and compared with other steganography methods (e.g., FNNS and LISO) in terms of compression rate, PSNR, and robustness.

**Weaknesses:**

1. Weak Motivation: The paper’s focus on JPEG compression as the main adversarial scenario for steganography is narrow. JPEG is not the sole or the most challenging attack vector in steganography, limiting the practical significance of this work.
2. The claim of "joint optimization of image compression and steganography" is not convincingly supported, as the paper mainly demonstrates standard compression techniques applied to a latent code without introducing novel optimization strategies. Integrating a GAN-based discriminator to improve image quality is not a new concept and has been widely used in image steganography. This reduces the paper's innovative contribution.
3. Confusing Experimental Comparisons: The experiments are difficult to follow, with unclear captions and a lack of structure in presenting results.
4. Lack of Diverse Baseline and Attack Comparisons: The evaluation would be more robust with additional baseline methods and attack scenarios beyond JPEG compression. Including comparisons with a broader range of steganography techniques and different attack types (e.g., scaling, noise addition) would provide a better understanding of the model’s resilience and comparative performance.
5. Parameter Choices and Absence of Ablation Study: The model’s reliance on multiple parameters (λ1 to λ5) without any ablation study to show their individual effects. A detailed ablation study would strengthen the paper by showing the impact of each parameter and justifying their choices.
6. The security claims would be stronger with evaluations using neural network-based steganalysis tools, as these are increasingly relevant in steganography. Including such tests would provide a more robust validation of the method's security.

**Questions:**

The approach is highly dependent on existing pre-trained models (ELIC and Hyperprior), which raises questions about generalizability and scalability. The paper lacks an exploration of how the method would perform with different compression models, limiting the flexibility and applicability of the approach.
Relying on specific models makes the method harder to adapt or extend to other compression or steganography frameworks. This reduces the scope of the method to limited model architectures, constraining its adoption.

---

### Official Review · Reviewer_qycn · 2024-11-04

**Soundness:** 2
**Presentation:** 2
**Contribution:** 2
**Rating:** 6
**Confidence:** 2

**Summary:**

This paper proposes a novel method for robust image steganography that withstands JPEG compression. Instead of directly modifying image pixels, the approach uses a semi-amortized optimization technique in the latent space to jointly optimize image compression and steganography. By combining popular learned compression models (hyperprior and ELIC) with a pre-trained steganography model (FNNS), the method achieves superior compression, high fidelity, and accurate decoding, outperforming existing methods like FNNS-JPEG and LISO-JPEG.

**Strengths:**

- This article is written clearly.

- This paper first proposes a joint optimization of compression and steganography, which solves the problem that steganographic images are degraded.

- The experimental results demonstrate attractive performance.

**Weaknesses:**

Could the authors provide an example of a neural network-based steganography method being practically applied? My research area is not in this field, but it seems to me that this field offers little practical value beyond papers and some handcraft metrics.

**Questions:**

See weakness

---

### Meta-Review · Area_Chair_kQsP · 2024-12-20

**Metareview:**

The motivation of this paper is weak, wherein the paper’s focus on JPEG compression as the main adversarial scenario for steganography is narrow. Integrating a GAN-based discriminator to improve image quality is not a new concept and has been widely used in image steganography. This reduces the paper's innovative contribution.
The experiment is confusing. The evaluation would be more robust with additional baseline methods and attack scenarios beyond JPEG compression.

**Additional Comments On Reviewer Discussion:**

Thanks a lot for all reviewers' hard working. As all our reviewers turn to reject this paper, my recommendation is "Reject".

---

### Decision · Program_Chairs · 2025-01-22

Reject